# Preparation and Characterization of Magnetic Fe_3_O_4_/CdWO_4_ and Fe_3_O_4_/CdWO_4_/PrVO_4_ Nanoparticles and Investigation of Their Photocatalytic and Anticancer Properties on PANC1 Cells

**DOI:** 10.3390/ma12193274

**Published:** 2019-10-08

**Authors:** Mohammad Amin Marsooli, Mahdi Fasihi-Ramandi, Kourosh Adib, Saeid Pourmasoud, Farhad Ahmadi, Mohammad Reza Ganjali, Ali Sobhani Nasab, Mahdi Rahimi Nasrabadi, Marta E. Plonska-Brzezinska

**Affiliations:** 1Nanobiotechnology Research Center, Baqiyatallah University of Medical Sciences, Tehran 1951683759, Iran; mohammadamin.marsuli@gmail.com; 2Faculty of Pharmacy, Baqiyatallah University of Medical Sciences, Tehran 6461853090, Iran; 3Molecular Biology Research Center, System Biology and Poisoning Institute, Baqiyatallah University of Medical Sciences, Tehran 1951683759, Iran; fasihi.m@gmail.com; 4Department of Chemistry, Imam Hossein University, Tehran 1955735345, Iran; k_anbaz@yahoo.com; 5Department of Physics, University of Kashan, Kashan 8731753153, Iran; SA_POURMASOUD2007@yahoo.com; 6Physiology Research Center, Iran University of Medical Sciences, Tehran 1449614535, Iran; Farhadahmadi55@gmail.com; 7Department of Medicinal Chemistry, School of Pharmacy-International Campus, Iran University of Medical Sciences, Tehran 1451555763, Iran; 8Center of Excellence in Electrochemistry, Faculty of Chemistry, University of Tehran, Tehran 1951683759, Iran; Ganjali@gmail.com; 9Biosensor Research Centre, Endocrinology & Metabolism Molecular and Cellular Research Institute, Tehran University of Medical Sciences, Tehran 1951683759, Iran; 10Social Determinants of Health (SDH) Research Center, Kashan University of Medical Sciences, Kashan 8115187159, Iran; Ali.sobhaninasab@gmail.com; 11Core Research Lab, Kashan University of Medical Sciences, Kashan 8115187159, Iran; 12Department of Organic Chemistry, Faculty of Pharmacy with the Division of Laboratory Medicine, Medical University of Bialystok, Mickiewicza 2A, 15-222 Bialystok, Poland

**Keywords:** magnetic nanoparticle, Fe_3_O_4_/CdWO_4_, Fe_3_O_4_/CdWO_4_/PrVO_4_, sonochemical procedure, photocatalysis, methylene blue, cytotoxicity properties, MTT assay, human cell line, PANC1 cells

## Abstract

Fe_3_O_4_/CdWO_4_ and Fe_3_O_4_/CdWO_4_/PrVO_4_ magnetic nanoparticles were prepared at different molar ratios of PrVO_4_ to previous layers (Fe_3_O_4_/CdWO_4_) via the co-precipitation method assisted by a sonochemical procedure, in order to investigate the photocatalytic performance of these systems and their cytotoxicity properties. The physico-chemical properties of these magnetic nanoparticles were determined via several experimental methods: X-ray diffraction, energy dispersive X-ray spectroscopy, Fourier transformation infrared spectroscopy and ultraviolet-visible diffuse reflection spectroscopy, using a vibrating sample magnetometer and a scanning electron microscope. The average sizes of these nanoparticles were found to be in the range of 60–100 nm. The photocatalytic efficiency of the prepared nanostructures was measured by methylene blue degradation under visible light (assisted by H_2_O_2_). The magnetic nanosystem with a 1:2:1 ratio of three oxide components showed the best performance by the degradation of ca. 70% after 120 min of exposure to visible light irradiation. Afterwards, this sample was used for the photodegradation of methyl orange, methyl violet, fenitrothion, and rhodamine-B pollutants. Finally, the mechanism of the photocatalytic reaction was examined by releasing ^•^OH under UV light in a system including terephthalic acid, as well as O^2−^, OH, and hole scavengers. Additionally, the cytotoxicity of each synthesized sample was assessed using a 3-(4,5-dimethylthiazol-2-yl)-2,5-diphenyltetrazolium bromide assay against the human cell line PANC1 (cancer), and its IC50 was approximately 125 mg/L.

## 1. Introduction

Obtaining aseptic conditions for several commercial products is the main factor associated with public health problems. In recent years, several researchers have attempted to discover and develop new strategies against organic pollutants [1,2,3]. The cosmetics, leather, paper, plastics, and textile industries, among others, produce many pollutants, including colour contaminants, that are carcinogenic to mammals due to their use of dyestuffs and chemical materials [4,5]. Nanoparticles that are effective for combining various original properties in a single nanoplatform have received considerable attention [1,4,5]. Currently, one of the most investigated tools to eliminate harmful contaminants from water are photocatalytic materials [6,7,8,9,10,11,12,13,14,15,16,17,18,19]. Magnetic nanoparticles act as heterogeneous catalysts that can easily separate a catalyst from a reaction system with the aid of an added magnetic field [19,20,21,22]. Furthermore, the use of H_2_O_2_ to improve photocatalytic results under visible and UV light has been reported in previous studies [23,24]. 

Semiconductor materials such as ZnS, TiO_2_, WO_3_, and CdS are promising candidates for the degradation of contaminated environments via photocatalytic reactions [25,26]. For example, transition metal-based orthovanadates (MVO_4_) such as PrVO_4_ have the potential to be applied in several fields, including the gas-sensor, solar-cell, catalyst, photoluminescent, and photocatalyst fields [27,28,29,30,31]. 

CdWO_4_ and some of its composites have shown considerable potential for the photocatalytic degradation of some of the colours under visible and UV irradiation [32,33,34]. CdWO_4_ showed its great potential in its physicochemical behavior resulting from the self-trapped excitons [32,35]. CdWO_4_, which crystallizes in a monoclinic wolframite structure, has substantial chemical, optical, and structural properties [32,36]. Its energy band gap is bigger than common photocatalysts like TiO_2_, suggesting that it would have a low photo-absorption activity [23,24]. Nonetheless, CdWO_4_ is able to be excited via a considerable amount of energy which is higher than its band gap, generating high energy electron-hole pairs. Therefore, this compound has a considerable photocatalytic activity for removing several contaminants and dyes from water, resulting in turning poisonous pollutants into safe compounds such as H_2_O as well as CO_2_ [20,37]. However, the quick recombination ratios of generated charge carriers as well as the inadequate response to visible rays can decrease the usage of CdWO_4_. The photocatalytic productivity of rare earth elements doped with CdWO_4_ is bigger than that of undoped samples for degrading organic pollutants [1,10,38]. 

Cancer is an important public health threat affecting global disability and mortality rates. Pancreatic cancer shows a high incidence worldwide. It has been shown that new therapeutic approaches are needed to complement the current therapeutic tools. Targeting cancer cells is one of the most important uses of nanoscale-designed materials [39,40]. Nanoparticles are very promising since they can perform as drug carriers and sono- and photosensitizers [41,42,43]. In these last two cases, the formation of reactive oxygen species is observed as a result of electromagnetic radiation excitation with ultrasound activation or an adequate wavelength, respectively [41,43]. Due to the interesting physicochemical properties, excellent stability under physiological conditions and high biocompatibility at a low material complexity of inorganic nanoparticles, they may be used in cancer therapy. Since then, the most frequently used have been titanium dioxide [41,42,44], zinc oxide [41], and tungstate nanoparticles [45,46,47,48]. 

This study reports the synthesis of Fe_3_O_4_/CdWO_4_ (signed as S1) and Fe_3_O_4_/CdWO_4_/PrVO_4_ magnetic nanocomposites with varying molar ratios of PrVO_4_ (1:2:0.5 (S2); 1:2:0.75 (S3); 1:2:1 (S4); and 1:2:1.5 (S5)) with an enhanced photocatalytic activity under visible light irradiation. The morphological, structural, magnetic, optical and cytotoxic properties of the optimized sample were obtained using scanning electron microscopy (SEM), X-ray diffraction (XRD), an energy dispersive analysis of X-rays (EDX), Fourier transformation infrared spectroscopy (FT-IR), vibrating sample magnetometry (VSM), and UV-Vis diffuse reflection spectroscopy (DRS). Two types of pollutants, methylene blue (MB) and methyl violet (MV), were used to evaluate the photocatalytic activity of the samples under visible light irradiation (assisted by H_2_O_2_). Additionally, in order to assess the efficiency of the synthesized samples under ultraviolet light, the samples with the contaminants MB, fenitrothion (FNT), methyl orange (MO), and rhodamine-B (RhB) were tested. The photocatalytic behaviour mechanism was investigated using the conversion test of terephthalic acid (TA) to 2-hydroxy-TA in the presence of hydroxyl radicals and the employment of ^•^O^2−^, ^•^OH, and hole scavengers. Additionally, the 3-(4,5-dimethylthiazol-2-yl)-2,5-diphenyltetrazolium bromide (MTT) assay was performed on a PANC1 cell line on S4 nanoparticles to calculate the cytotoxicity effect on mammalian cancer cells.

## 2. Materials and Methods 

### 2.1. Materials

Unless noted, all chemicals and solvents were commercially available and used as received without further purification from Sigma, Germany. We used iron (III) chloride hexahydrate (FeCl_3_·6H_2_O) (99%), iron(II) chloride tetrahydrate (FeCl_2_·4H_2_O) (99%), sodium hydroxide (NaOH) (98%), cadmium nitrate tetrahydrate (Cd(NO_3_)_2_·4H_2_O) (99%), sodium tungstate dehydrate (Na_2_WO_4_·2H_2_O) (99%), praseodymium (III) nitrate hexahydrate (Pr(NO_3_)_3_·6H_2_O) (99%), ammonium metavanadate (NH_4_VO_3_) (99%), methyl orange (MO) (99%), methylene blue (MB) (99%), methyl violet (MV) (99%), fenitrothion (FNT) (99%), rhodamine B (RhB) (99%), foetal bovine serum (FBS) (99%), penicillin (99%), streptomycin (99%), trypsin (99%), 2,2′,2′′,2′′′-(ethane-1,2-diyldinitrilo) tetraacetic acid (EDTA) (99%), 3-(4,5-dimethylthiazol-2-yl)-2,5-diphenyltetrazolium bromide (MTT) (99%), acetone (99.8%) and ethanol (99.8%). All aqueous solutions were made using deionized water, which was further purified with a Milli-Q system (Millipore). 

#### 2.1.1. Preparation of Fe_3_O_4_ Nanoparticles

First, 0.01 moles of FeCl_3_·6H_2_O and 0.005 moles of FeCl_2_·4H_2_O were dissolved in 200 mL of distilled water and transferred to a three-neck flask [49]. A mechanical stirrer was used for stirring the solution for 60 min. Then, by adding 2 M NaOH at 30 °C in the presence of gaseous N_2_, the precipitation process was carried out. The reaction system was kept at 70 °C for 5 h, and the pH of the solution was ± 12. Afterward, the system was cooled to room temperature, and the precipitation was separated via a permanent magnet and washed with distilled water until the pH was 7. Finally, the Fe_3_O_4_ nanoparticles were washed with acetone and dried at 70 °C. 

#### 2.1.2. Preparation of Fe_3_O_4_/CdWO_4_ Nanoparticles

First, 0.001 moles of Fe_3_O_4_ nanoparticles were dispersed by ultrasonication in 50 mL of distilled water. Then, 50 mL of Cd(NO_3_)_2_·4H_2_O (0.04 M solution) was added to the container containing the Fe_3_O_4_ nanoparticles. Afterward, 50 mL of Na_2_WO_4_·2H_2_O (0.04 M solution) was added dropwise to the mixture, while a magnetic mixer stirred the contents at room temperature. The precipitation was separated by the magnet, washed with distilled water and ethanol, and dried at 80 °C. Sediments were placed at 550 °C for 3 h for calcinations.

#### 2.1.3. Preparation of Fe_3_O_4_/CdWO_4_/PrVO_4_ Nanoparticles

In situ co-precipitation (assisted by the ultrasonic approach) was used for the preparation of these nanoparticles. First, 0.544 g of noncalcified as-prepared Fe_3_O_4_/CdWO_4_ (sample S1) was dispersed in 70 mL of distilled water under ultrasonic irradiation for 15 min. Then, in two separate containers, 50 mL of 0.01 M NH_4_VO_3_ and Pr(NO_3_)_3_·6H_2_O were prepared to obtain sediments with a molar ratio of 1:2:0.5 (sample S2), and the NH_4_VO_3_ solution was added to the dispersed nanoparticles. The ultrasonic probe was placed in the mixture for the generation of ultrasonic waves with a frequency and power of 20 KHz and 400 W, respectively. The contents of the Pr(NO_3_)_3_·6H_2_O solution were added dropwise to the reaction vessel for 5 min and then exposed to ultrasound irradiation for 15 min. To fabricate sediments with various molar ratios, quantities of 0.087 g NH_4_VO_3_ and 0.325 g Pr(NO_3_)_3_·6H_2_O (with molar ratios of 1:2:0.75, sample S3), 0.117 g NH_4_VO_3_ and 0.44 g Pr(NO_3_)_3_·6H_2_O (1:2:1, sample S4), as well as 0.175 g NH_4_VO_3_ and 0.651 g Pr(NO_3_)_3_·6H_2_O (1:2:1.5, sample S5) were used in the same approach. The sediments were separated with a magnet, washed with distilled water and ethanol, and then dried in the oven at 75 °C. Finally, the synthesized nanoparticles were placed at 550 °C for 3 h.

To achieve a more accurate understanding of the synthesis of nanocomposites, the proposed mechanism is expressed as follow:(1)$$\mathrm{Fe}{\left({\mathrm{NO}}_{3}\right)}_{3}\cdot 9H_{2}O+H_{2}O\to {\;\mathrm{Fe}}^{3+}\;+\;3{{\mathrm{NO}}_{3}}^{-}\;+\;10H_{2}O$$
(2)$$\mathrm{Fe}{\left({\mathrm{NO}}_{3}\right)}_{2}\cdot 9H_{2}O+H_{2}O\to {\;\mathrm{Fe}}^{2+}\;+\;2{{\mathrm{NO}}_{3}}^{-}\;+\;10H_{2}O$$
(3)$${\mathrm{Fe}}^{2+}+{\;\mathrm{Fe}}^{3+}+5\mathrm{NaOH}+5{{\mathrm{NO}}_{3}}^{-}+H_{2}O\to \;\mathrm{Fe}{\left(\mathrm{OH}\right)}_{2}\;+\;\mathrm{Fe}{\left(\mathrm{OH}\right)}_{3}+\;5{\mathrm{NaNO}}_{3}$$
(4)$$\mathrm{Fe}{\left(\mathrm{OH}\right)}_{2}\;+\;2\mathrm{Fe}{\left(\mathrm{OH}\right)}_{3}+\;\Delta \;\to {\;\mathrm{Fe}}_{3}O_{4}+\;4H_{2}O$$
(5)$$\mathrm{Cd}{\left({\mathrm{NO}}_{3}\right)}_{2}\cdot 6H_{2}O+H_{2}O\;\to {\;\mathrm{Cd}}^{2+}\;+\;2{{\mathrm{NO}}_{3}}^{-}\;+\;7H_{2}O$$
(6)$${\mathrm{Na}}_{2}{\mathrm{WO}}_{4}\cdot \;2H_{2}O+{\;H}_{2}O\;\to \;2{\mathrm{Na}}^{+}+{\;\mathrm{WO}}_{4}^{2-}\;+\;3H_{2}O$$
(7)$${\mathrm{Cd}}^{2+}+{\;\mathrm{WO}}_{4}^{2-}\to {\;\mathrm{CdWO}}_{4}$$
(8)$${\mathrm{NH}}_{4}{}^{+}\;+\;{{\mathrm{NO}}_{3}}^{-}\to {\;\mathrm{NH}}_{4}{\mathrm{NO}}_{3}$$
(9)$$\mathrm{Pr}{\left({\mathrm{NO}}_{3}\right)}_{3}\cdot 6H_{2}O+H_{2}O\to {\mathrm{Pr}}^{3+}\;+\;3{{\mathrm{NO}}_{3}}^{-}\;+\;7H_{2}O$$
(10)$${\mathrm{NH}}_{4}{\mathrm{VO}}_{3}\;+{\;H}_{2}O\;\to {\mathrm{NH}}_{4}{}^{+}\;+{{\mathrm{VO}}_{3}}^{-}+{\;H}_{2}O$$
(11)$${\mathrm{Pr}}^{3+}+{\;\mathrm{VO}}_{4}^{3-}\to {\;\mathrm{PrVO}}_{4}$$
(12)$$\begin{array}{c} \mathrm{Pr}{\left({\mathrm{NO}}_{3}\right)}_{3}\cdot 6H_{2}O\;+3{\mathrm{NH}}_{4}{\mathrm{VO}}_{3}+\mathrm{Fe}{\left({\mathrm{NO}}_{3}\right)}_{3}\cdot 9H_{2}O\;+\mathrm{Fe}{\left({\mathrm{NO}}_{3}\right)}_{2}\cdot 9H_{2}O+\;\mathrm{Cd}{\left({\mathrm{NO}}_{3}\right)}_{2}\\ \cdot 6H_{2}O+\;2{\mathrm{Na}}_{2}{\mathrm{WO}}_{4}\;+\;\mathrm{NaOH}\;\\ \to {\;\mathrm{NH}}_{4}{\mathrm{NO}}_{3}+{\mathrm{NaNO}}_{3}+{\;\mathrm{Fe}}_{3}O_{4}/{\mathrm{CdWO}}_{4}/{\mathrm{PrVO}}_{4}\;+H_{2}O \end{array}$$

### 2.2. Methods

#### 2.2.1. Assessment of Photocatalytic Performance

To identify the optimal sample, the photocatalytic performance of all the synthesized samples was assessed by MB photodegradation under visible light. In each experiment, 60 mg of the dispersed photocatalyst was added to 300 mL of 25 ppm MB solution. Then, 1 mL of 25% H_2_O_2_ was added to the photoreactor. Previous to exposure to the visible spectrum (250 W xenon lamp), the solution was stirred in darkness for 20 min to reach an adsorption/desorption equilibrium between the catalyst and the MB solution. Then, 4 mL of the solution was kept in darkness for 10 min and then for 20 min under the light. Next, the solutions were centrifuged at 5000 rpm for 5 min to separate the catalysts. A UV-Vis spectrophotometer was used to determine the outcome of the photodegradation of MB. Finally, the photocatalytic activity of the S4 sample was determined via the photodegradation of 10 mg/L of MO and 10 mg/L of MV.

The photodegradation of MB under UV light was evaluated. In each experiment, 30 mg of the photocatalyst was added to 300 mL of 20 mg/L MB solution. Before irradiation under UV light (50 W Hg lamp), the solution was stirred in the dark for 20 min to get an adsorption/desorption equilibrium between the catalyst and the MB solution. Then, 4 mL of the solution was pipetted every 10 min and centrifuged at 5000 rpm for 5 min to separate the catalyst. The concentration of MB solution was calculated via a UV-Vis spectrophotometer to identify the result of the photodegradation. Afterward, the optimized sample was used for the photodegradation of 15 mg/L MO, 15 mg/L FNT, and 20 mg/L RhB via the same method.

#### 2.2.2. Photodegradation Mechanism

Hydroxyl radicals (^•^OH) produced at the photocatalyst/water interface were analysed via terephthalic acid (TA) as a probe via a photoluminescence technique. A high fluorescence intensity of 2-hydroxyterephthalic acid is achieved by TA with ^•^OH. Therefore, the intensity of fluorescence is directly proportional to the concentration of ^•^OH. The experiments were similar to the photocatalytic testing under ultraviolet irradiation. The reaction was performed as follows: 0.03 g of photocatalyst (0.1 g/L) was added into the 300 mL aqueous solution of terephthalic acid with a concentration of 0.0005 M (0.451 g in 0.5 L distilled water) along with 0.002 M NaOH (0.04 g in 0.5 L distilled water). The principal oxidative species in the photocatalytic procedure were obtained, respectively, via the super oxide radical (^•^O^2−^), ^•^OH, and holes, using benzoquinone, tert-butanol, and citric acid. First, 300 mL of 25 mg/L MB and 3 mmol from one of the scavengers was added to the solution. Next, 0.03 g of a dispersed photocatalyst was subjected to the ultraviolet irradiation. Finally, 4 mL of each solution was centrifuged, and the process was monitored through a UV-Vis spectrophotometer.

#### 2.2.3. Cell Culture

PANC1 cell lines were prepared from the National Cell Bank of Iran (NCBI, Tehran). The cell line was grown in RPMI 1640 medium (Gibco) and supplemented with 10% (*v*/*v*) FBS and penicillin/streptomycin (100 IU/mL and 100 μg/mL, respectively). The cells were incubated and preserved at 37 °C with 5% CO_2_. As soon as confluence reached ca. 85%, the cells were rinsed with pure RPMI and gathered using a 0.25% trypsin/EDTA solution. Each test was performed 3 times.

#### 2.2.4. MTT Assay

An MTT assay was used to assess the cytotoxicity of the extract on the PANC1 cells. The potential of viable cells was determined via the production of a blue formazon catalyst from yellow tetrazolium salt through mitochondrial dehydrogenase. The cells were collected and plated in a 96-well plate (Nunc, Denmark) at a density of 104 cells/well and were treated with varying concentrations of nanoparticles (2, 1, 0.5, 0.25, 0.125, 0.063, 0.0315, and 0.0157 mg/mL). For 1 and 2 days, the microplates were incubated at 37 °C and 5% CO_2_. Then, the supernatants were discarded, and 100 μL of DMSO was added to each well and further incubated for 20 min. The ELISA plate reader was used at λ = 570 nm. The percentage of cell cytotoxicity and viability was achieved using the following relation [29]:(13)$$cytotoxicity\;\left(\%\right)=1-\;\frac{mean\;absorbance\;of\;toxicant}{mean\;absorbance\;of\;negative\;control}\;\times \;100.$$

## 3. Results and Discussion

### 3.1. Characterization of Synthesized Nanostructures

The X-ray diffraction (XRD) patterns of the powder materials of Fe_3_O_4_, Fe_3_O_4_/CdWO_4_ and the Fe_3_O_4_/CdWO_4_/PrVO_4_ sample (S4) are shown in Appendix A, and the main reflections are summarized in Appendix A. As seen in Appendix A, the Fe_3_O_4_ nanoparticles showed six peaks at the 2θ degrees of 30.0°, 35.6°, 42.9°, 53.5°, 56.9° and 63.1°, which correspond to the cubic phases of Fe_3_O_4_ (JCPDS No. 01-075-0449) [50]. These peaks are related to the reflection of the X-rays from the (220), (311), (400), (411), (511), and (440) lattice planes, respectively. The average crystallite diameter (Dc), also calculated, was 14.1 nm (Appendix A). The XRD pattern of Fe_3_O_4_/CdWO_4_ (Appendix A) is composed of two pure phases of Fe_3_O_4_ (JCPDS No. 01-75-0449) and CdWO_4_ (JCPDS No. 01-084-1457) with diffraction peaks characteristic of the cubic phase of Fe_3_O_4_, as mentioned above, with additional peaks at 2θ = 23.2°, 29°, 35.2°, 40°, 47.5°, 50.1°, 50.3°, 51.11° and 60°, with the lines (110), (111), (002), (210), (112), (030), (022), (130) and (041), respectively. The Dc value was calculated as ca. 33.3 nm (Appendix A). The XRD pattern of the Fe_3_O_4_/CdWO_4_/PrVO_4_ nanostructures consists of reflections from three phases of Fe_3_O_4_ (JCPDS No. 01-075-0449), CdWO_4_ (JCPDS No. 01-084-1457) and PrVO_4_ (JCPDS No. 084-1457) (Appendix A). Appendix A shows diffraction peaks at 24.03° (line (200)), 32.04° (line (112)), 40.1° (line (231)), 47.86° (line (322)) and 50.26° (line (111)). The Dc value of the Fe_3_O_4_/CdWO_4_/PrVO_4_ nanostructures was evaluated, and it was ca. 55 nm. The reflections from the PrVO_4_ planes are very low in intensity because of the small amount of this inorganic oxide present in the nanoparticles in comparison to other inorganic components. No other impurities or additional peaks were observed, which confirmed the high purity of the synthesized products. 

As illustrated in Appendix A, the EDX spectrum of the Fe_3_O_4_/CdWO_4_/PrVO_4_ (sample S4) nanoparticles is made up of six varied elements, Fe, O, Cd, W, Pr, and V. Furthermore, no impurity peaks were found, indicating that the Fe_3_O_4_/CdWO_4_/PrVO_4_ nanoparticles have a high purity. 

The surface morphologies of the Fe_3_O_4_/CdWO_4_ and Fe_3_O_4_/CdWO_4_/PrVO_4_ nanoparticles (samples S1 and S4) were studied using scanning electron microscopy (SEM). The structures of S1 and S4 exhibit porous morphologies, with numerous channels and outcroppings (Figure 1a,b). All of the samples are composed of aggregates of the nanoparticles with sizes under 100 nm. The Fe_3_O_4_/CdWO_4_ nanoparticles formed aggregates with an average size of 60–70 nm (Figure 1c). The S4 sample shows a uniform morphology with larger crystallites in the range of 90–100 nm (Figure 1d). In addition, the outcomes proved that the particle size obtained by SEM is larger than that obtained by XRD. This refers to the point where the SEM images show the aggregates of many inorganic crystallites. 

The particle sizes of the synthesized nanoparticles were also estimated from the high-resolution transmission electron microscopy (HRTEM) (Figure 1e,f). All of the HRTEM images show spherical Fe_3_O_4_/CdWO_4_ and Fe_3_O_4_/CdWO_4_/PrVO_4_ nanoparticles with mean particle sizes of 20–30 (Figure 1e) and 50–55 nm (Figure 1f), respectively. The PrVO_4_ coating of the Fe_3_O_4_/CdWO_4_ process leads to the formation of some polygonal and spherical nanostructures with different diameters (Figure 1f). 

In Figure 2, the magnetic behaviour of the nanosized structures was verified via the hysteric curve at 300 K and the nearly saturated nature. The results showed the magnetic contribution of the as-fabricated Fe_3_O_4_/CdWO_4_/PrVO_4_ nanoparticles at room temperature. Furthermore, the VSM data validated that the as-fabricated products could be classified as paramagnetic nanomaterials, and their magnetization values were approximately 51 and 0.13 emu/g for the Fe_3_O_4_ and Fe_3_O_4_/CdWO_4_/PrVO_4_ nanoparticles at room temperature, respectively.

The band gap energies of the prepared nanoparticles were examined via UV-Vis absorption spectroscopy, as shown in Figure 3. From the UV-Vis spectra (Figure 3a), the band gap was found by extrapolating the steepest portion of the (αhν)^1/2^ vs. hν plot by using Tauc’s formula:(14)$$\alpha hv=A{\left(hv-E_{g}\right)}^{\eta }.$$

Tauc’s plots were made for the Fe_3_O_4_/CdWO_4_ (S1) and Fe_3_O_4_/CdWO_4_/PrVO_4_ (S4) samples gaps of the material (Figure 3b). Using Equation (14), the energy gaps were calculated for the S1 and S4 samples and were determined to be 3.1 and 2.8 eV, respectively.

To identify the functional groups and oxide metal bonding in Fe_3_O_4_, Fe_3_O_4_/CdWO_4_ (sample S1) and Fe_3_O_4_/CdWO_4_/PrVO_4_ (sample S4) before and after the calcination process, and after photocatalysis in the presence of the MB dye, Fourier transformation infrared (FT-IR) spectroscopy was used; the results are presented in Figure 4. The FT-IR spectra were recorded between the 450 cm^−1^ and 3500 cm^−1^ wavelengths at room temperature. 

The absorption bands in the range of 1618–1629 cm^−1^ and 3390–3410 cm^−1^, observed for all spectra, are assigned to the deformation vibration of the H–O–H bonds and to the stretching vibration of the O–H bonds [35], respectively (Figure 4). The absorption band at 587 cm^−1^ is related to the Fe–O vibration (Figure 4a) [51]. Some additional absorption bands at 455, 716, 819, and 893 cm^−1^ were observed for the CdWO_4_ nanostructures, and they are attributed to the vibration modes present in these nanoparticles after the absorption of the infrared wavelength (Figure 4b). The peak at 455 cm^−1^ may be attributed to the Cd–O stretching vibration mode, whereas the peaks at 716 and 819 cm^−1^ were due to O–W–O, and the peak at 820 cm^−1^ was due to Cd–O–W [36,37,52,53]. The FT-IR spectra confirm the presence of stretching and bending vibrations of metal cations, such as the Cd–O, O–W–O and Cd–O–W bands in the CdWO_4_ structure. 

Figure 4c,d shows the FT-IR spectra of the S4 sample before and after calcination, respectively. The peak at 816 cm^−1^ (Figure 4c), which was replaced by 827 cm^−1^ (Figure 4d), is related to the calcination process. The peak became more intense and may be related to the vibrational modes of the V–O bond. The small absorption peak at 451 cm^−1^ belongs to the Pr–O vibration frequency [54]. Figure 6e shows the FT-IR spectra of the S4 sample after the photodegradation of MB by UV light irradiation. No change in the absorption spectra of the inorganic nanoparticles was observed, which indicates that these nanoparticles are stable and were not destroyed during the photocatalytic reactions. 

### 3.2. Photocatalytic Performance

The photocatalytic properties of the synthesized samples, assisted by 100 mL MB in 1 mL H_2_O_2_, under visible light irradiation, were tested to find the nanoparticles with the highest photocatalytic performance. The efficiency of H_2_O_2_, a non-catalytic pollutant photodegradation, was also studied with and without light to evaluate the photocatalytic properties of the synthesized nanoparticles. The results are shown in Figure 5a. Additionally, the kinetics of the photocatalytic processes in terms of the irradiation time were determined (−ln(C/C_0_)). The mathematical analysis is presented in Figure 5b. The slope of the linear regression was utilized as the first-order reaction rate constant. A comparison of all samples clearly showed (Figure 5) that the higher photocatalytic activity was performed by S4 (Fe_3_O_4_/CdWO_4_/PrVO_4_). S4 has a good potential to eliminate all organic contaminants. 

Similar experiments on the photodegradation of MV and MO were performed with the same sample (S4), and the results are displayed in Figure 6. Some additional photocatalytic degradation tests were carried out using MB, MO, FNT, and RhB pollutants to gain a better understanding of the S4 properties under an ultraviolet wavelength, and the results are shown in Figure 7. Therefore, the reduced size of sample S4 led to an increase in the surface of the nanoparticles as well as more absorption under ultraviolet rays, improving the production of the radical species and resulting in the enhancement of the degradation of dyes.

The photocatalytic degradation of pollutants occurs via the reactive sample, after the light absorption and the electron-hole formation by the photocatalyst [55]. The terephthalic acid (TA) photoluminescence technique was used to study the generation of active ^•^OH radicals for all samples, as summarized in Appendix A [38]. By means of ^•^OH by TA, 2-hydroxyl-TA could be formed. This has a high fluorescence radiation, and as a result the ^•^OH could be monitored incidentally once we monitored the changes in the fluorescence intensity of the TA solution. The change in the fluorescence intensity of 2-hydroxyl-TA is shown in Appendix A. Therefore, any increase in ^•^OH can lead to an increase in 2-hydroxyl-TA, which has a fluorescence property. Hence, the production of ^•^OH radicals will be improved, and as a result, the degradation of the dyes will be enhanced.

As shown in Appendix A, the UV irradiation time is directly proportional to the fluorescence intensity. However, the intensity of ^•^OH reaches its minimum value during the first 10 min in the absence of irradiation. This indicates the formation of ^•^OH all over the photocatalyst with ultraviolet waves. Due to the factors described in the absence of UV light, we obtained a minimal amount of ^•^OH. Conversely, in the presence of UV light, the fluorescence was intensified; this was interpreted as a larger production of ^•^OH, which led to an increase in the degradation of the dyes.

Trapping tests of holes (h+) were used to establish the principal oxidative samples via superoxide radical (O_2_*^•^*^−^) and *^•^*OH, citric acid, benzoquinone, as well as tert-butanol [10]. As shown in the picture, citric acid can remove holes (h+) in the solution, which results in the degradation of the dyes by up to 92%. Therefore, holes (h+) have no significant effect on the solution. However, without O_2_*^•^*^−^ or *^•^*OH, the degradation of the dyes was observed at levels of 35% and 76%, respectively. These findings suggest that both radicals (O_2_*^•^*^−^ and *^•^*OH) could be considered to be important reactive species in the photocatalytic destruction reaction of MB in the presence of Fe_3_O_4_/CdWO_4_/PrVO_4_.
Fe_3_O_4_/CdWO_4_/PrVO_4_ + hν → Fe_3_O_4_/CdWO_4_/PrVO_4_^•^ + e^−^ + h^+^(15)
e^−^ + O_2_ → O_2_**^•-^**(16)
O_2_**^•-^** + H_2_O → OOH^•^ + OH**^-^**(17)
OOH^•^ → O_2_ + H_2_O_2_(18)
H_2_O_2_ + O_2_**^•-^** → OOH^•^ + OH**^-^** + O_2_(19)
OH^•^ + MB→ destruction products(20)
O_2_**^•-^** + MB→ destruction products(21)

The outcomes of applying these scavengers for the photodegradation of MB (25 ppm) are represented in Figure 8. The addition of a superoxide scavenger to the studied solution could lead to a reduction of the photocatalytic performance of inorganic nanoparticles by up to two thirds. Interestingly, the addition of a hydroxyl scavenger resulted in a noticeable decrease in the photocatalytic performance (less than half). However, the addition of a holes scavenger had an insubstantial influence on the photocatalytic activity of the inorganic nanoparticles. 

### 3.3. Cytotoxicity Effect on PANC1 Cells

The MTT assay shows that the S4 nanocomposite had a toxic effect on a PANC1 cell line in a dose-depended manner, and its IC50 was approximately 125 mg/L (Figure 9). Additionally, Appendix A presents the microscopic photos of PANC1 cells with S4 at the three different concentrations. The presented studies confirmed that the toxicity of the S4 sample was reduced by reducing its concentration. The in vitro studies indicated that the Fe_3_O_4_/CdWO_4_/PrVO_4_ nanoparticles were able to inhibit the growth of the PANC1 cancer cells. Therefore, these inorganic nanoparticles have some potential to be developed as new and novel anticancer agents for the treatment of pancreatic cancer based on the outcome provided as primary evidence [56].

## 4. Conclusions

In summary, magnetic Fe_3_O_4_/CdWO_4_ and Fe_3_O_4_/CdWO_4_/PrVO_4_ nanostructures were prepared at various molar ratios of inorganic salts via the co-precipitation method assisted by the ultrasonic technique. XRD, EDS, SEM, and FTIR methods established the presence of the desired nanoparticles with different transition metals. The DRS data showed an important reduction in the band gap when adding PrVO_4_ to the “core” phases (Fe_3_O_4_ and CdWO_4_). The DRS test showed that the OBGEs of Fe_3_O_4_ and the S4 sample were 3.1 and 2.8 eV, respectively. The VSM test determined the MS values of Fe_3_O_4_ and S4, which were 50.9 and 0.13 emu/g, respectively. The highest photocatalytic activity was shown by Fe_3_O_4_/CdWO_4_/PrVO_4_, with a ratio of 1:2:1 (S4). The degradation of MB with a 70% yield under visible light was observed. This sample was also used for the photodegradation of MO, MV, FNT, and RhB under visible and UV light. The IC50 of the S4 sample on a PANC1 cell line was approximately 125 mg/L, as determined by the MTT assay. 

## Figures and Tables

**Figure 1 materials-12-03274-f001:**
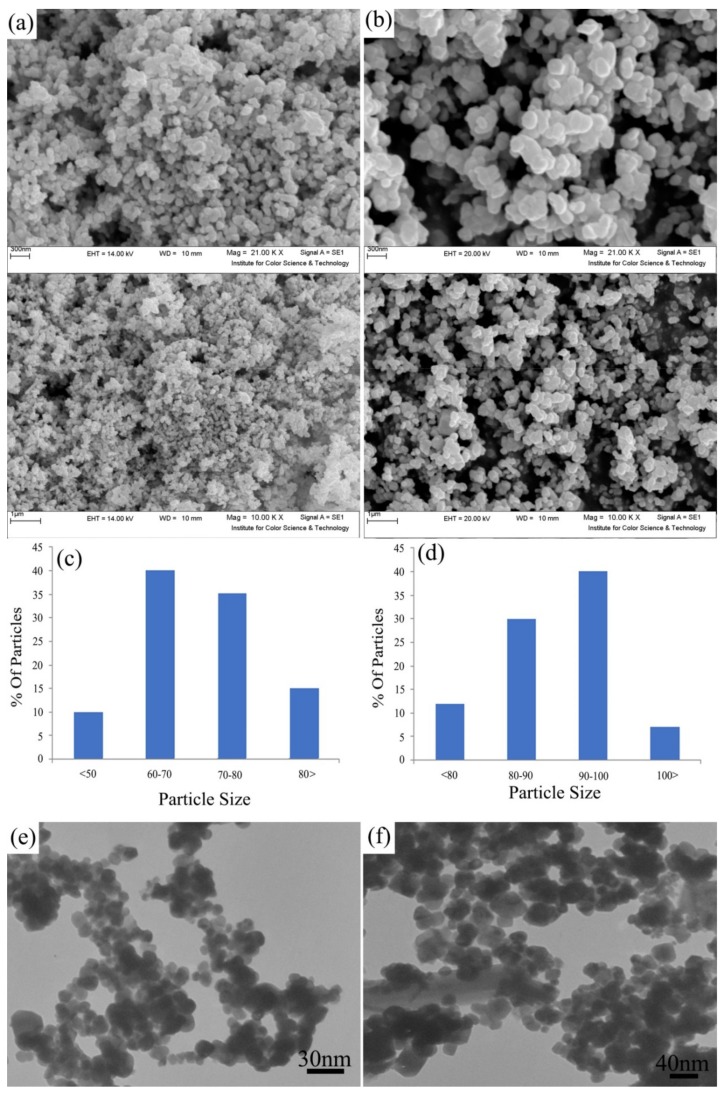
(**a**,**b**) The SEM images, (**c**,**d**) particle size distribution and (**e**,**f**) HRTEM images of the (**a,c,e**) Fe_3_O_4_/CdWO_4_ (S1) and (**b,d,f**) Fe_3_O_4_/CdWO_4_/PrVO_4_ (S4) samples.

**Figure 2 materials-12-03274-f002:**
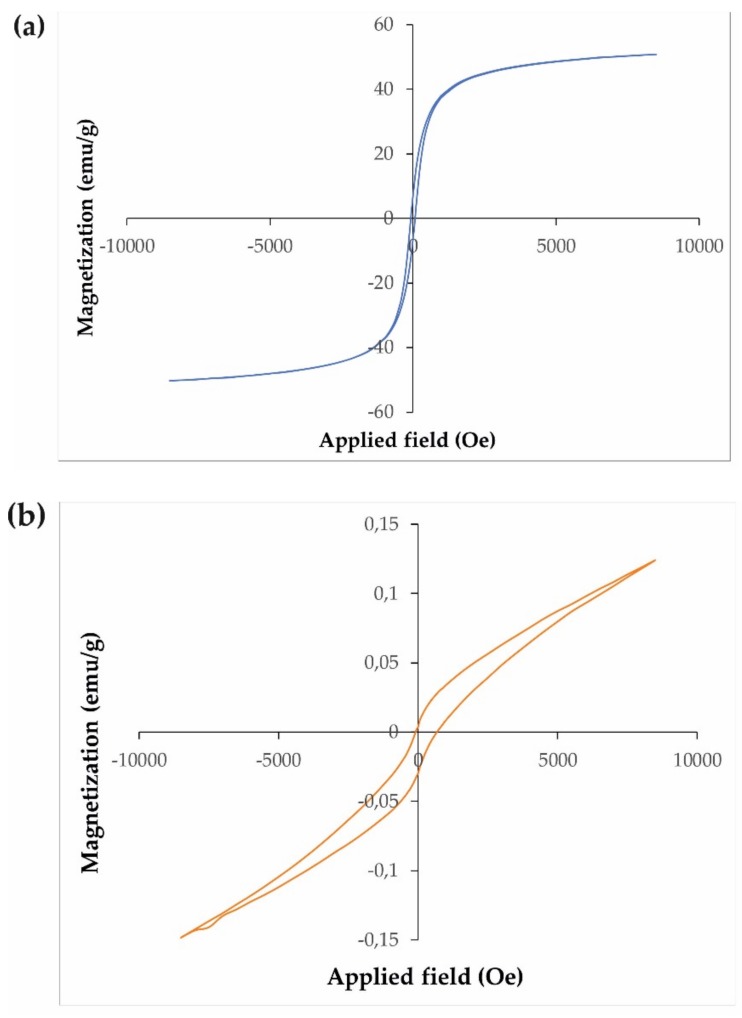
The magnetic hysteresis loops of the (**a**) Fe_3_O_4_ and (**b**) Fe_3_O_4_/CdWO_4_/PrVO_4_ (S4) samples.

**Figure 3 materials-12-03274-f003:**
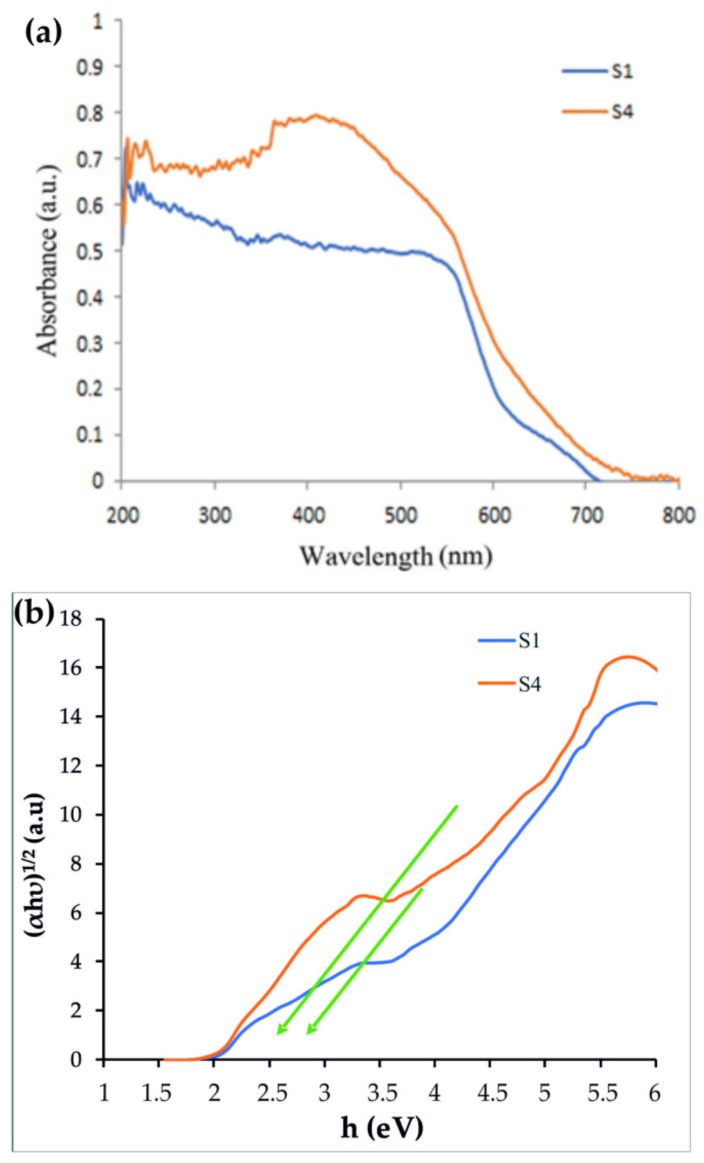
The (**a**) UV-vis absorbance spectra and (**b**) Tauc’s plots for the Fe_3_O_4_/CdWO_4_ (S1) and Fe_3_O_4_/CdWO_4_/PrVO_4_ (S4) samples.

**Figure 4 materials-12-03274-f004:**
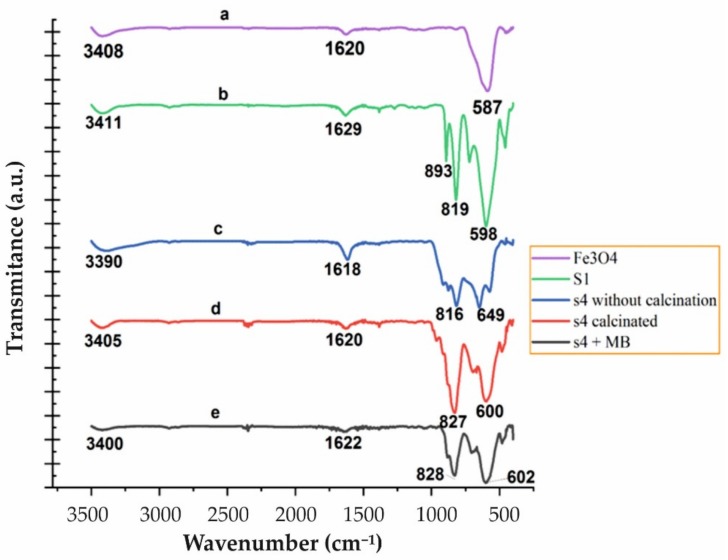
The FTIR spectra of (**a**) Fe_3_O_4_, (**b**) Fe_3_O_4_/CdWO_4_ (S1), and (**c**) Fe_3_O_4_/CdWO_4_/PrVO_4_ (S4) before calcination, (**d**) Fe_3_O_4_/CdWO_4_/PrVO_4_ (S4) after calcination, and (**e**) Fe_3_O_4_/CdWO_4_/PrVO_4_ (S4) after a period of MB photodegradation.

**Figure 5 materials-12-03274-f005:**
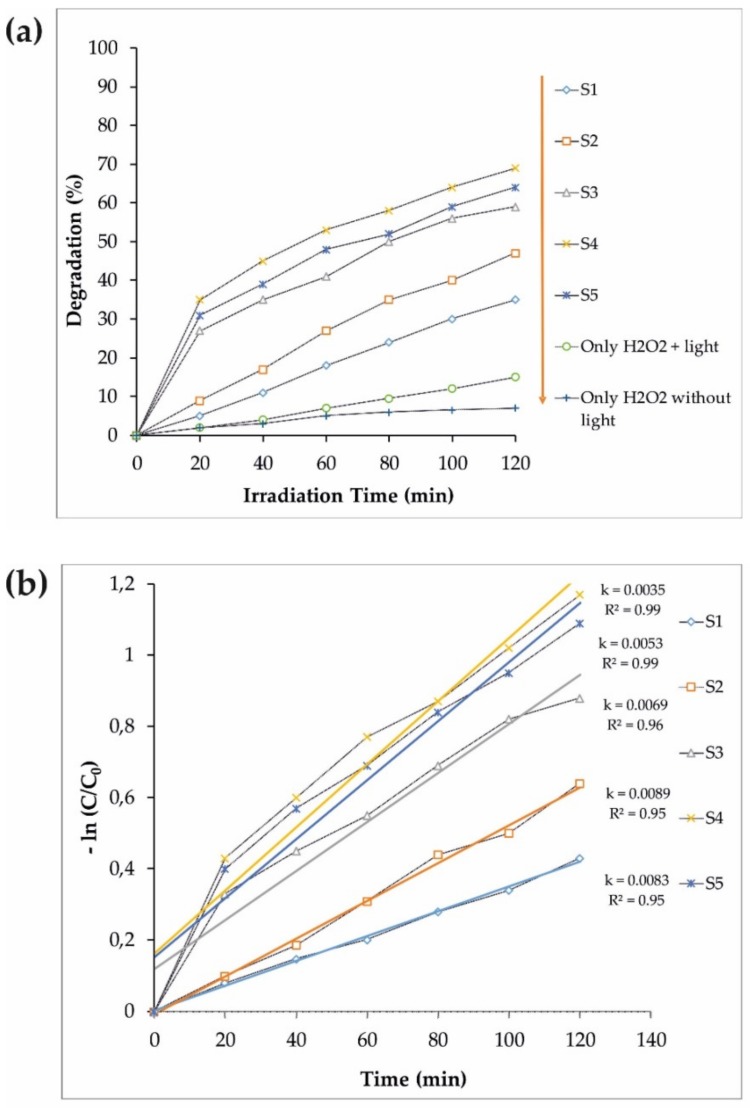
(**a**) The photocatalytic degradation of MB (S1–S5 samples) under visible light assisted by H_2_O_2_. (**b**) The pseudo-first-order kinetics of the MB degradation for S1–S5 samples.

**Figure 6 materials-12-03274-f006:**
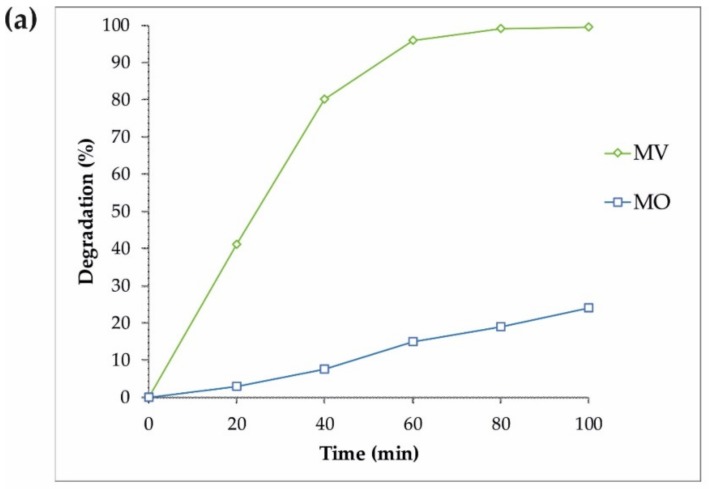
(**a**) The photocatalytic degradation of the MV and MO pollutants by the S4 sample under visible light assisted by H_2_O_2_, and (**b**) the pseudo-first-order kinetics of the MV and MO degradation by S4.

**Figure 7 materials-12-03274-f007:**
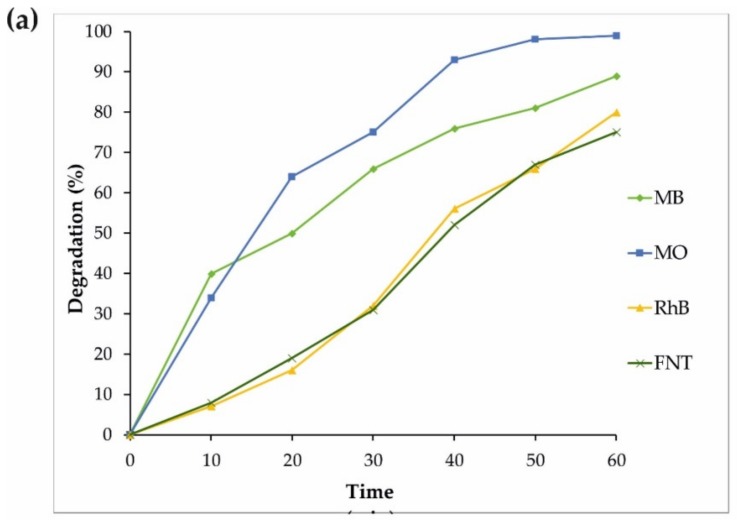
(**a**) The photocatalytic degradation of MB, MO, RhB, and FNT by the S4 sample under UV light irradiation. (**b**) The pseudo-first-order kinetics of the MB, MO, FNT, and RhB degradation by the S4 sample.

**Figure 8 materials-12-03274-f008:**
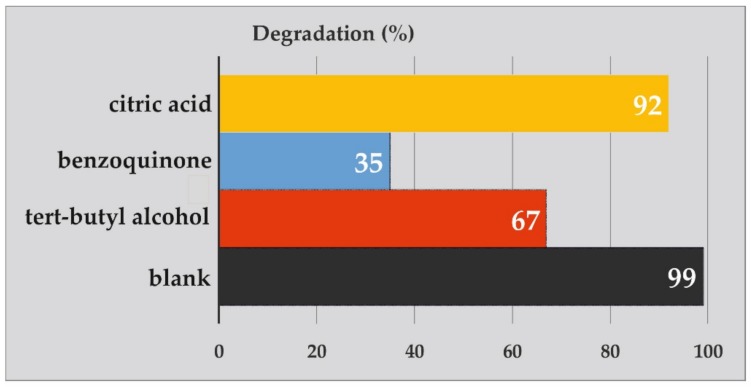
The effect of different scavengers (benzoquinone, tert-butyl alcohol and citric acid) on the photocatalytic degradation of MB.

**Figure 9 materials-12-03274-f009:**
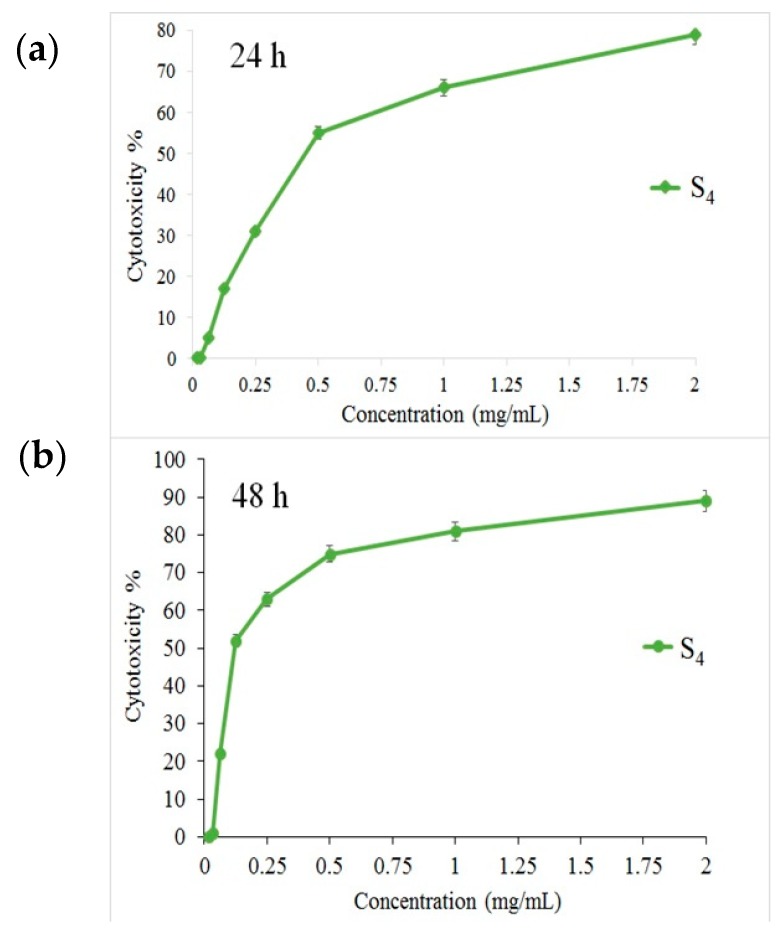
The relative in vitro cell viability of optimized Fe_3_O_4_/CdWO_4_/PrVO_4_ (MTT assay). PANC1 cells incubated with Fe_3_O_4_/CdWO_4_/PrVO_4_ for (**a**) 24 h and (**b**) 48 h.

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
