# Peer review of "Preparation and Characterization of Magnetic Fe3O4/CdWO4 and Fe3O4/CdWO4/PrVO4 Nanoparticles and Investigation of Their Photocatalytic and Anticancer Properties on PANC1 Cells"

_materials, 2019, doi:10.3390/ma12193274_

Round 1
Reviewer 1 Report
This work describes the preparation and characterization of various hybrid magnetic nanoparticles for photocatalytic and anti-cancer properties. While this work is interesting, the following issues must be addressed before becoming suitable for publication:
The introduction is not a comprehensive overview of the literature and could be framed better to highlight the focus of the study. The author could dedicate the first paragraph to the advantages of magnetic nanoparticles and then divide the following paragraph into recent advances into photocatalytic nanoparticles and then the recent advances in anti-cancer nanoparticles, with focus on metal oxides and tungstates/vanadates. Merging the two applications in one paragraph makes the introduction seem a bit disorganized. After that, the provided rationale for using CdWO4 and other vanadates is suitable. Please consider including the following references in the introduction: Oleshkevich, E.; Teixidor, F.; Rosell, A.; Viñas, C. Merging Icosahedral Boron Clusters and Magnetic Nanoparticles: Aiming toward Multifunctional Nanohybrid Materials. Inorg. Chem. 2018, 57, 462−470 Ramachandran, R.; Jung, D.; Bernier, N. A.; Logan, J. K.; Waddington, M. A.; Spokoyny, A. M. Sonochemical Synthesis of Small Boron Oxide Nanoparticles. Inorg. Chem. 2018, 57, 8037−8041. Here are some errors in the reactions of the proposed synthesis: Please use accurate symbols to show the water of hydration in the chemical formulae for all equations. For example, Fe(NO3)39H2O should be written as Fe(NO3)3.9H2O The charges are not balanced in any of the chemical reactions. If you are going to represent the reactions in redox format, the charges have to be balanced as well to have a net charge of zero. Please refer to a standard general chemistry textbook for detailed information about balancing redox reactions. In equation 4, deltaT is included as a reactant. This is an incorrect method of representing heat as it actually represents change in temperature. Please represent heat as delta or write the temperature on top of the reaction arrow. The nanoparticles are characterized by PXRD which indicates the presence of crystalline phases. It would be beneficial to show elemental composition from the EDX analyses to give us accurate information about the ratio of both metals, especially in the mixed oxide nanoparticles. Considering that the nanoparticles are 50-70 nm in size, the SEM images do not accurately highlighting their size. Please provide a higher magnification SEM image of the nanoparticles along with a histogram depicting the number of particles versus their size in order to confirm this. TEM/HRTEM microscopy would be a great addition to the characterization in order to confirm the size and crystalline facets of the nanoparticles. For the images shown in Figure 9, please show the results of the appropriate control experiments in order to support your hypothesis. Without this information, it is not possible to conclude that the observed effect is due to the synthesized nanoparticles.Author Response
Reviewer #1: This work describes the preparation and characterization of various hybrid magnetic nanoparticles for photocatalytic and anti-cancer properties. While this work is interesting, the following issues must be addressed before becoming suitable for publication:
The introduction is not a comprehensive overview of the literature and could be framed better to highlight the focus of the study. The author could dedicate the first paragraph to the advantages of magnetic nanoparticles and then divide the following paragraph into recent advances into photocatalytic nanoparticles and then the recent advances in anti-cancer nanoparticles, with focus on metal oxides and tungstates/vanadates. Merging the two applications in one paragraph makes the introduction seem a bit disorganized. After that, the provided rationale for using CdWO4 and other vanadates is suitable. Please consider including the following references in the introduction: Oleshkevich, E.; Teixidor, F.; Rosell, A.; Viñas, C. Merging Icosahedral Boron Clusters and Magnetic Nanoparticles: Aiming toward Multifunctional Nanohybrid Materials. Inorg. Chem. 2018, 57, 462−470 Ramachandran, R.; Jung, D.; Bernier, N. A.; Logan, J. K.; Waddington, M. A.; Spokoyny, A. M. Sonochemical Synthesis of Small Boron Oxide Nanoparticles. Inorg. Chem. 2018, 57, 8037−8041.
Comment: We reorganized Introduction of the manuscript and added some requested informations and references. On Pages 2 and 3, we made some corrections, modified and expanded the text. Every changes are highlighted.
Reviewer #1: Here are some errors in the reactions of the proposed synthesis: Please use accurate symbols to show the water of hydration in the chemical formulae for all equations. For example, Fe(NO3)39H2O should be written as Fe(NO3)3.9H2O The charges are not balanced in any of the chemical reactions. If you are going to represent the reactions in redox format, the charges have to be balanced as well to have a net charge of zero. Please refer to a standard general chemistry textbook for detailed information about balancing redox reactions.
Comment: In response to this reviewer request, we did corrections and every changes are highlighted.
Reviewer #1: In equation 4, deltaT is included as a reactant. This is an incorrect method of representing heat as it actually represents change in temperature. Please represent heat as delta or write the temperature on top of the reaction arrow.
Comment: Corrected.
Reviewer #1: The nanoparticles are characterized by PXRD which indicates the presence of crystalline phases. It would be beneficial to show elemental composition from the EDX analyses to give us accurate information about the ratio of both metals, especially in the mixed oxide nanoparticles.
Comment: EDX analyses is shown in Figure S2 and some comments were added in the text on Page 6.
Reviewer #1: Considering that the nanoparticles are 50-70 nm in size, the SEM images do not accurately highlighting their size. Please provide a higher magnification SEM image of the nanoparticles along with a histogram depicting the number of particles versus their size in order to confirm this. TEM/HRTEM microscopy would be a great addition to the characterization in order to confirm the size and crystalline facets of the nanoparticles.
Comment: In response to this reviewer request, we provided additional SEM images with higher magnifications and histogram with the particles diameter. HRTEM images were also included. Additional experiments were added as Panels c-f in Figure 1.
Reviewer #1: For the images shown in Figure 9, please show the results of the appropriate control experiments in order to support your hypothesis. Without this information, it is not possible to conclude that the observed effect is due to the synthesized nanoparticles.
Comment: In response to this reviewer request, we replaced Figure S4 with Figure 9. In Figure 9 the results for the control experiments are included.

Reviewer 2 Report
Author have investigated detail study on the preparation and characterization of magnetic Fe3O4/CdWO4 and Fe3O4/CdWO4/PrVO4 nanoparticles and investigation of their photocatalytic and anticancer properties on PANC1 cells. However, I would like to suggest few comments to author to consider.
In the introduction part, the significance of Fe3O4/CdWO4 and Fe3O4/CdWO4/PrVO4 NPs must be improved. The UV-Visible absorption spectra of Fig. 3 should be be included in the revision. Fig. S3 must be include and compare with the analysis of all S1-S5 samples.
Author Response
Reviewer #2: Author have investigated detail study on the preparation and characterization of magnetic Fe3O4/CdWO4 and Fe3O4/CdWO4/PrVO4 nanoparticles and investigation of their photocatalytic and anticancer properties on PANC1 cells. However, I would like to suggest few comments to author to consider. In the introduction part, the significance of Fe3O4/CdWO4 and Fe3O4/CdWO4/PrVO4 NPs must be improved.
Comment: We reorganized Introduction of the manuscript and added some requested informations and references. On Pages 2 and 3, we made some corrections, modified and expanded the text.
Reviewer #2: The UV-Visible absorption spectra of Fig. 3 should be included in the revision.
Comment: In response to this reviewer request, we included in our manuscript UV-Visible absorption spectra in Figure 3 as a Panel (a).
Reviewer #2: Fig. S3 must be include and compare with the analysis of all S1-S5 samples.
Comment: The results for all samples are summarized in Table S2.

Round 2
Reviewer 1 Report
The authors have addressed the reviewer comments and the manuscript is much improved as a result. I can now recommend the manuscript for publication.
Reviewer 2 Report
Author response to reviewer comments are satisfactory. I recommend that the revised manuscript can be accepted for publication.